# Spatially Mapping the Baseline and Bisphenol-A Exposed *Daphnia magna* Lipidome Using Desorption Electrospray Ionization—Mass Spectrometry

**DOI:** 10.3390/metabo12010033

**Published:** 2022-01-01

**Authors:** Matthew J. Smith, Ralf J. M. Weber, Mark R. Viant

**Affiliations:** School of Biosciences, University of Birmingham, Edgbaston, Birmingham B15 2TT, UK; m.smith.15@bham.ac.uk (M.J.S.); r.j.weber@bham.ac.uk (R.J.M.W.)

**Keywords:** *Daphnia magna*, lipidome, desorption electrospray ionization, mass spectrometry imaging, ion images, bisphenol-A, egg development, cryosectioning

## Abstract

Untargeted lipidomics has previously been applied to the study of daphnids and the discovery of biomarkers that are indicative of toxicity. Typically, liquid chromatography—mass spectrometry is used to measure the changes in lipid abundance in whole-body homogenates of daphnids, each only ca. 3 mm in length which limits any biochemical interpretation of site-specific toxicity. Here, we applied mass spectrometry imaging of *Daphnia magna* to combine untargeted lipidomics with spatial resolution to map the molecular perturbations to defined anatomical regions. A desorption electrospray ionization—mass spectrometry (DESI-MS) method was optimized and applied to tissue sections of daphnids exposed to bisphenol-A (BPA) compared to unexposed controls, generating an untargeted mass spectrum at each pixel (35 µm^2^/pixel) within each section. First, unique lipid profiles from distinct tissue types were identified in whole-body daphnids using principal component analysis, specifically distinguishing appendages, eggs, eye, and gut. Second, changes in the lipidome were mapped over four stages of normal egg development and then the effect of BPA exposure on the egg lipidome was characterized. The primary perturbations to the lipidome were annotated as triacylglycerides and phosphatidylcholine, and the distributions of the individual lipid species within these classes were visualized in whole-body *D. magna* sections as ion images. Using an optimized DESI-MS workflow, the first ion images of *D. magna* tissue sections were generated, mapping both their baseline and BPA-perturbed lipidomes.

## 1. Introduction

*Daphnia magna* is a key sentinel species for environmental toxicology owing to its position in freshwater food webs as one of the most voracious primary consumers of phytoplankton. Consequently, changes to *Daphnia* fitness can induce significant perturbations to higher trophic levels. Additionally, this species is amenable to laboratory toxicity testing due to their rapid life cycle, asexual reproduction, and large numbers of offspring that are produced per adult. For these reasons, *D. magna* is utilized in the international OECD Guidelines for the Testing of Chemicals for both acute (OECD Test No. 202: *Daphnia* sp. Acute Immobilization [1]) and chronic exposures (OECD Test No. 211 *Daphnia magna* Reproduction Test [2]). However, these established tests are limited to measuring whole organism toxicity endpoints—specifically, survival and the number of offspring that are produced, respectively, making them information-sparse assays. Currently, there are calls to improve the adequacy of chemical test methods, including to measure more mechanistic data [3,4]. Adverse outcome pathways (AOPs) provide a knowledgebase for developing new chemical test methods that will focus on the detection of molecular perturbations that are predictive of adverse effects, i.e., toxicity endpoints [5]. The core of an AOP that is relevant to environmental toxicology comprises of early response molecular biomarkers that are indicative of adverse outcomes in larger biological structures up to an entire ecosystem. As such, well-developed AOPs should provide a rich mechanistic biochemical understanding of toxicity pathways, yet this requires the discovery, identification, and validation of early response molecular biomarkers that can predict downstream ecological effects.

Untargeted metabolomics is an approach in which analysts can detect changes in the abundance of metabolites in response to a stressor, most commonly using mass spectrometric analysis [6,7]. This has enabled the identification of early response molecular biomarkers in *D. magna* for a range of environmentally toxic chemicals. The oxidative stress that is caused by copper [8], obesogenic effects of tributyltin [9], neuroactive chemicals (i.e., nicotine and caffeine) [10], perfluorooctanesulfonic acid [11], and the impact that the endocrine disruptor bisphenol-A has on lipid storage and energy metabolism [12,13] have all been investigated. However, due to the analytical sensitivity of mass spectrometry, these studies required the homogenization of multiple whole daphnids during sample preparation to generate sufficient tissue mass per sample. Therefore, these studies are limited to providing an averaged metabolic (here meaning polar metabolic and/or lipid) phenotype of all the tissue types in *Daphnia*. While this approach benefits from biological averaging of multiple individuals, i.e., it reduces the metabolic variability across biological replicates, several limitations are apparent: first, a whole-body homogenate is unable to reveal site-specific toxicity. Also, by including tissues that are undergoing rapid metabolic changes, such as developing eggs, the whole organism metabolic phenotype can become dominated by such natural changes, potentially masking the subtle effects of toxicants. Thus, for an improved understanding of the metabolic consequences of toxicity in *D. magna*, spatial imaging of metabolism becomes necessary.

Mass spectrometry imaging (MSI) is a technology that ionizes chemical species at a user-defined spatial resolution that is directly from a sample’s surface, prior to mass spectrometry (MS) analysis to generate a mass spectrum at each pixel [14]. This should enable the study of distinct anatomical regions within individual whole-body *D. magna* sections, in principle allowing sites of toxicity to be identified and, through the use of metabolomics, the ability to measure a wide range of metabolic perturbations in specific tissues. There are many MSI technologies with matrix-assisted laser desorption ionization (MALDI) the most widely used for biological applications [15,16,17]. For optimal spatial resolution of a limited *m*/*z* range, secondary ion mass spectrometry (SIMS) is routinely used [18], and for the detection of a range of biomolecules (with lower spatial resolution) both liquid extraction surface analysis (LESA) [19,20] and desorption electrospray ionization (DESI) mass spectrometry are most appropriate. DESI is a non-destructive ambient ionization source that uses electrosprayed solvent ions to desorb, ionize, and ultimately direct analytes towards the MS detector [21]. It, therefore, requires minimal sample preparation compared to MALDI, offers spatial resolution as low as 25 um^2^, and enables multiple complementary experiments from a single sample, e.g., haematoxylin and eosin (H&E)-staining for improved integration of spatial information and orthogonal MSI methods to enhance the metabolome coverage [22].

MSI approaches have been broadly adopted in biomedical research [14,16,17], phytochemical analysis [23], and for investigating model organisms such as zebrafish and drosophila [24,25,26]. Specifically, Pirro et al. used DESI-MS to monitor lipid dynamics during the development of zebrafish embryos [25] and Liu et al. analyzed the changes in the phospholipid distribution in adult zebrafish upon exposure to fibronil, discovering that the perturbations were most evident in the eye region [27]. While not a study of endogenous metabolism, Böhme et al. reported the use of MSI to spatially map the uptake of engineered nanoparticles in *D. magna* [28]. Additionally, MALDI-MS has been conducted on whole *D. magna* that are directly deposited onto slides and cut into several pieces for high-throughput analysis of endogenous lipids [29] but no mass spectrometry study to date has reported spatial information on the endogenous metabolism of this small (ca. 3 mm in length) sentinel species.

The overall aim of this study was to demonstrate the capabilities of DESI mass spectrometry imaging that is applied to the small organism *D. magna* towards the spatial mapping of the baseline lipidome and its perturbations in response to chemical exposure. The first objective was to adapt the current best practices for histology, DESI-MS, and data processing to enable the first spatially-resolved MS measurements of *D. magna* to map lipids. Second, to spatially characterize the baseline lipidome of *D. magna*, seeking to identify the differences between the distinct anatomical regions (eggs, eye, gut, and appendages). The third objective was to characterize the baseline lipidome of *D. magna* eggs through their development (four time points, from 8 to 72 h). Lastly, we sought to investigate the effects of a model toxicant—bisphenol-A (BPA)—on the egg lipidome over the same time course. BPA is a precursor of polycarbondate plastics and epoxy resins, which is a known endocrine disruptor that affects the reproductive success [30]. It was selected as the model toxicant that was based on previously reported MS-detectable lipid perturbations in whole-body homogenates of *D. magna* that were indicative of reproductive toxicity [31].

## 2. Results

### 2.1. Application of Desorption Electrospray Ionization—Mass Specrometry to Daphnia magna

#### 2.1.1. Preparation of Tissue Sections

The first step to achieve spatially-resolved (toxico-) lipidomics data was to generate 20 µm thick cryosections from individual daphnids (Figure 1a—optical microscope image) such that the distinct anatomical regions could be distinguished by DESI-MS (with a pixel size that was set in the instrument software to 35 µm^2^). To maximize the likelihood of achieving sufficiently high quality cryosections, we chose to study adult *D. magna* (three- to four-week-old) to maximize the tissue size.

The established best practice for cryosectioning typically uses an operating cutting temperature (OCT) media to embed the samples, however, this is incompatible with MSI analysis since it contains polyethylene glycol (PEG)—a substance that is known to readily contaminate mass spectrometers [32]. Instead, most of the literature describing MS-compatible embedding media focuses on formulations of gelatin, carboxymethyl cellulose (CMC), and agarose hydrogels, owing to their cost effectiveness, availability, and physical properties [32,33,34,35]. Whilst these formulations have been previously evaluated [33], *D. magna* presents a unique challenge due to its small size and exoskeleton, so determining the optimal embedding media for cryosectioning this organism was necessary. This optimization was based on three criteria: (i) the physical state at ambient temperature, (ii) the physical state at the cutting temperature (−18 °C), and (iii) the quality of the tissue sections that were produced. These studies revealed that 1% CMC + 9% gelatin offered the optimal balance between fluidity at ambient temperature for a good coverage of the daphnids and the ability to maintain its structure during sectioning (at −18 °C), i.e., to support this delicate sample type (Appendix A).

Another important step in this process was the rapid cooling of the daphnids as soon as they were covered in the embedding media (at ambient temperature). This freezing step needed to be sufficiently rapid to limit ice propagation and cracking. To find the most suitable cryogen for snap-freezing, we compared liquid nitrogen (LN_2_), LN_2_ cooled isopentane, and liquid propane [36]. The evaluation of these cryogens used a quantitative scoring system that was based on distinguishing the anatomical regions from the H&E-stained sections that were prepared; this comparison of the three coolants was conducted with both OCT and our preferred MS-compatible embedding medium (1% CMC + 9% gelatin). The scoring (see Appendix A) revealed that LN_2_ outperformed the other cryogens and, given its relative ease of use, it was selected as our preferred freezing method. These scores showed no significant difference (two-sided *t*-test, *p* = 0.151) between the quality of the OCT embedded *D. magna* sections (the gold standard in histology) and those using our MS-compatible media, thus giving confidence in our MS-compatible tissue section preparation method. A representative H&E-stained section that was prepared using the optimized cryosectioning method is shown as an optical image in Figure 1b; many of the key anatomical regions within a daphnid are readily visible.

#### 2.1.2. Quality Control Strategies to Maximize DESI-MS Image Quality

To maximize the data quality from the spatially localized analysis of the *D. magna* lipidome, we needed to implement suitable quality control (QC) strategies to minimize *m*/*z* drift, signal decay, and background noise features in the mass spectra. This was particularly important as these unwanted effects tend to scale with analysis time; for example, we calculated that to achieve high spatial resolution (i.e., a pixel size of 35 µm^2^) with DESI-MS, each tissue section would require ca. five hours of data acquisition, hence requiring ca. nine days of acquisition time for the 24-sample study that is described below.

The QC strategies that were employed for this spatial lipidomic experiment were largely adapted from metabolomics best practices [6]. To monitor the *m*/*z* drift, rhodamine 6G dye was regularly analyzed over time [37] since it yielded a single intense peak of theoretical accurate mass *m*/*z* 443.2335. When the *m*/*z* error exceeded 25 ppm, we analyzed a film of polyalanine containing a range of known ions across the entire *m*/*z* range (Appendix A) to re-calibrate the mass spectrometer. The signal decay over time was monitored by regularly analysing a readily available and well-studied tissue section—porcine liver—with more similar surface and matrix effects to the *D. magna* sections. We re-optimized the DESI parameters once the detected lipid profile (Appendix A) intensity dropped below 1 × 10^5^ counts [37].

Finally, solvent clusters and other background ions were removed from the dataset using data that were acquired from a blank DESI slide at the start and end of data acquisition of the study samples. Any remaining temporal effects during the data acquisition (of technical origin) were minimized by block randomising the acquisition order of the *D. magna* tissue sections.

#### 2.1.3. Signal Processing the Study Samples to Yield Reliable Spatial Information

To process the lipidomic data that were generated by DESI-MS, we used the open source R package Cardinal [38]. Imaging-specific as well as routine LC-MS metabolomics data processing steps to (i) remove background pixels and noise features, (ii) group *m*/*z* values from the same analyte across the pixels of all the tissue sections in the study, and (iii) correct the technical variance in the intensity measurements were all considered and adapted as necessary. Diverging from the routine LC-MS metabolomics data processing approaches was necessary as metabolomics workflows tend to be optimized for handling similar mass spectra, which was not the case for the DESI data from imaging the whole-body *D. magna* sections, where each tissue section contained multiple distinct metabolic phenotypes. For instance, probabilistic quotient normalization and *m*/*z* feature filters that were set at high percentages were found to be unsuitable.

Our final signal processing workflow is split into four distinct parts, which we describe here with a rationale for each step. First, we extracted only the pixels that were pertaining to the *D. magna* tissue within the rectangular analysis area, hence removing many background pixels (Appendix A). This was important for the subsequent signal processing thresholds which were based on the assumption that all the data were from biological sources to ensure that each tissue section was subject to the same effective filtering (e.g., if 1000 background pixels were included when processing 1000 biological pixels, this would reduce the effective pixel filter by half, see below). We then processed the collection of mass spectra from the pixels within the individual tissue sections using: (i) a signal-noise ratio (SNR) filter to remove low intensity noise features with SNR < 3 based on the well-established LOD in analytical chemistry [39], (ii) *m*/*z* alignment within a 25 ppm tolerance that was derived from monitoring the *m*/*z* drift throughout the tissue section, (iii) a pixel filter to remove non-reproducible noise features that occurred in <5% of all the pixels in the tissue section, and (iv) root mean squared (RMS) normalization. Next, we aligned the spectra across all pixels in the study using a slightly less strict 35 ppm tolerance to account for the larger *m*/*z* drift over the multi-day study, and removed the intense background ions that were detected in the blank DESI slide. These processing steps achieved a reduction in the file sizes (per tissue section) from an initial 105 GB of raw study data to a more manageable ca. 8.5 GB, from which we were able to map the ion distributions for many hundreds of ions, with representative ion images shown in Figure 1c,d for *m*/*z* 756.5521 and 861.6503, respectively. Finally, the data were processed to address the specific biological objectives by (i) extracting data from the specific *D. magna* regions-of-interest (ROIs), (ii) removing anomalous pixels, and (iii) applying strict pixel and missing value filters, yielding data matrices for single feature statistical analysis and, following missing value imputation and pareto scaling, for principal component analysis (PCA)—as discussed in Section 2.2, Section 2.3 and Section 2.4. Due to its relative complexity with multiple steps that were optimized during this study, the entire workflow is summarized in Figure 2 for the convenience of the reader.

### 2.2. Lipidomic Differences between Daphnia magna Tissue Types

Having optimized the workflows, the initial biological investigation that was conducted was to investigate any differences in the baseline lipidomes of the distinct anatomical regions in *D. magna*, specifically between egg, eye, gut, and the appendages. This was conducted firstly as a technical check to ensure our DESI mass spectrometry imaging workflow could measure and distinguish what we believed would be major lipidomic differences in the daphnids. Additionally, it enabled the characterization of the baseline lipidomes of some important tissues within these animals upon which the subsequent toxicology study would be anchored.

A PCA was applied to the processed DESI-MS dataset that consisted of pixels only from the four selected anatomical regions. Some separation between the different tissue types was observed in the scores plot (Figure 3), giving confidence that the quality of the DESI-MS data, from the workflow that was outlined in Section 2.1, is suitable for biological investigations. To gain insights into the underlying lipidomic differences between these tissues, the PCA loadings plot was inspected (Appendix A).

The lipid classes that were predominantly driving the separation of the tissue types in this unsupervised multivariate analysis were the phosphatidylcholines and triacylglycerides. Boxplots and the corresponding ion images of two representative lipid species, one from each lipid class—specifically PC(34:03) and TG(50:06) where we used the standard naming format for lipids describing the number of carbon atoms and double bonds—are shown in Figure 4a,b, respectively, indicating their localization within *D. magna*. These results revealed the accumulation of phosphatidylcholines in the developing eggs and eye, and triacylglycerides that were predominantly stored in the appendages and eggs.

### 2.3. Temporal Effects in the Baseline Lipidome during Egg Development

Next, we investigated the baseline lipidome of *D. magna* eggs over time—specifically the development of the eggs over the seventh adult instar (i.e., the egg development phase between the sixth and seventh brood)—sampling three replicate daphnids at 8, 24, 48, and 72 h after the sixth brood. To uncover the temporal effects, we processed and analysed pixels from only the eggs in *D. magna* sections.

A PCA was applied to the processed dataset from the unexposed *D. magna* eggs, revealing a clear temporal effect in the scores plot (Figure 5). The largest source of variance in this DESI-MS dataset, accounting for 68% of the total variation, can be attributed to egg development, with the low PC1 values corresponding to early development (eight hours after the sixth brood) and the high PC1 corresponding to mature eggs immediately prior to release (72 h after the sixth brood). Pixels from the 24 and 48 h tissue sections are less separated in the PCA scores plot but their PC1 values lie between the 8 and 72 h time points. The PCA also shows that the biological variation between eggs in the same brood chamber and eggs from different daphnids increased over time.

The changes in the lipidome through egg development were dominated by phosphatidylcholines (high PC1) and triacylglycerides (low PC1), shown by the loadings plot in Appendix A. The change in abundance of a representative phosphatidylcholine and triacylglyceride are shown in Figure 6. The distribution of each lipid species across the whole daphnid over the four time points, are visualized with overlaid ion images that are set to a common intensity scale. These results suggest there was an increase in PC(32:00) and a decrease in TG(48:05) abundance in both the egg lipidome and in the female daphnid’s storage sites (as outlined in Section 2.2).

### 2.4. Perturbation of the Daphnia magna Egg Lipidome to Bisphenol-A

In our final analyses, we investigated the lipid perturbation in developing eggs upon acute exposure to the chemical stressor BPA. The sample set included the same time points as in Section 2.3 and a single BPA exposure concentration, with each group comprising of three replicate daphnids. The toxicity range-finding experiments to determine this exposure concentration were conducted using an adapted version of the OECD Test No. 202: *Daphnia* sp. Acute immobilization test—starting immediately after the sixth brood—to ensure that the animals were of sufficient size for DESI-MS. Our range-finding revealed an EC50 of 14 mg/kg (Appendix A), which matched well with the 12.8 mg/kg that was reported in an earlier study of the effects of BPA on *D. magna* neonates [40]. We selected an exposure concentration of 5 mg/kg BPA to minimize the lethality whilst maximising the likelihood of inducing measurable molecular effects by DESI-MS [31].

To evaluate the effects of BPA on egg development, we extracted the egg pixels from each tissue section (as described in Section 2.3), then subset the data into each of the four time points to determine the effect of the BPA relative to the age-matched control samples. A subsequent PCA of the egg pixels at each time point indicated that the largest effects of BPA on *D. magna* eggs occurred at 48 h, as shown in the PCA scores plots in Figure 7 and Appendix A. We thus focused on the egg pixels from the 48 h DESI-MS dataset to better understand the lipidomic effects of BPA exposure. The PCA scores plot (Figure 7) showed that 69% of the variance related to the effect of BPA exposure, however much of this variance also occurred within the control egg pixels across PC1 and PC2 both within the individual daphnids and between different animals (Appendix A).

The detected lipid perturbations upon BPA exposure were predominantly changes to phosphatidylcholines (high PC1) and triacylglycerides (low PC1)—evident through the loadings plot in Appendix A. The changes in the levels of the representative phosphatidylcholine and triacylglyceride lipid species between the BPA-exposed and the control egg pixels over the seventh adult instar are summarized in Figure 8. These findings show that the abundance of PC(32:00) generally increased in the control eggs over time, but decreased at 48 h upon BPA exposure before reaching comparable levels to the controls immediately before the release of neonates. In contrast, the abundance of TG(50:06) gradually decreased in the control eggs over time, but increased at 48 h into egg development upon BPA exposure before reaching a similar concentration to the controls at 72 h. The changes in the abundance of these lipids at 48 h were tested using a Wilcoxon signed-rank test, yielding significant FDR adjusted *p*-values of 1.2414 × 10^22^ for PC(32:00) and 1.97476 × 10^22^ for TG(50:06). Figure 8 also includes the ion images of PC(32:00) and TG(50:06) in the control and BPA-exposed daphnid sections to highlight the changes in their distribution in the developing eggs upon exposure as described in this section, but also across the whole adult daphnid. Mapping the lipids across the whole animal indicated that the lipidome changes in the eggs were mirrored in the main localization sites of the female daphnid (as outlined in Section 2.2); specifically, the abundance of phosphatidylcholines and triacylglycerides decreased and increased, respectively, at these storage sites after 48 h of BPA exposure.

Attempts were made to detect BPA in the exposed tissue sections to map the toxicant’s distribution, however, this was not achieved, indicating that the concentration of BPA in the sections were below the limit of detection of our DESI-MS set-up.

## 3. Discussion

### 3.1. Lipidomic Differences between Daphnia magna Tissue Types

Multiple studies have investigated the lipidome of *Daphnia magna* using mass spectrometry approaches, but, in this work, we report for the first time the spatial mapping of endogenous lipids in this sentinel species. DESI-MS offers a suitable spatial resolution for MSI of *D. magna*, allowing important anatomical regions to be distinguished. Additionally, since DESI-MS data acquisition time scales with the square of the pixel size, attempting to increase the spatial resolution would incur a significant time penalty, hence there is a trade-off between the size of a toxicity study, image resolution, and the acquisition time. Intuitively, a proposed benefit of higher spatial resolution would be the ability to image smaller *D. magna*, i.e., neonates, within a similar time frame; however we believe this would be at the cost of a vastly increased complexity in sample handling and preparation, as well as lower quality sections with currently accessible technologies.

We believe the QC strategies that were employed here ensured that the methods were optimized until suitably high-quality data were generated to gain new biological understanding of the *Daphnia* lipidome. Of note is the absence of an ‘intrastudy QC’ sample type which is highly important in routine LC-MS metabolomics for multiple reasons (feature filtering, assessing drift across analytical runs, measuring technical variation etc. [6]), yet due to the fundamental differences in sample preparation between MSI and LC-MS metabolomics, generating a single section that is representative of the entire sample set (which is the traditional method for preparing an intrastudy QC) was not feasible.

Throughout our analyses (Section 2.2, Section 2.3 and Section 2.4), we discovered that the lipidomic changes were dominated by the phosphatidylcholine and triacylglyceride lipid classes. This can be rationalized both technically and biologically as follows: in our studies, DESI-MS has limited lipidome coverage of ca. 250 lipids after all of the filtering steps (compared to routine LC-ESI-MS lipidomics which typically measures several thousands of features [8]) and the phosphatidylcholines and triacylglycerides were some of the most abundant and readily ionizable of the lipids with the specific DESI-MS set-up; additionally, phosphatidylcholines and triacylglycerides are key lipid classes that are involved in egg development and the responses to BPA exposure [31]. We suggest that both of these factors contributed to these two lipid classes dominating the analyses.

To increase coverage of the daphnid lipidome, complementary imaging data could be acquired from the same tissue sections. For example, further DESI-MS experiments could be conducted using both ionization polarities, different solvent compositions, and/or capillary temperatures. In addition, a single orthogonal MSI technique such as MALDI could be applied, as has recently been achieved by Zemaitis et al. for the study of rat brain tissue [41]. This takes advantage of the non-destructive nature of DESI and would be especially beneficial in studies aiming, for example, to additionally characterize the exposure chemical (and its biotransformation products) with spatial resolution.

A key challenge with untargeted MSI-based lipidomics is the identification of lipids with a high enough confidence for biological inference. Unlike for LC-MS lipidomics, it is less common to generate orthogonal analytical data (i.e., retention time and MS/MS data to facilitate lipid identification) given the time, cost, and practical considerations. However, recent advances in both instrumentation and computation, such as METASPACE [42], cyclicIM-MSI methodologies for lipid ID [43], and ozone-induced dissociation-MSI [44] open the door to that prospect. In this study, the lipid annotations are putative and are based on experimental MS1 data [45]. To minimize the likelihood of false positive annotations, the MS1 data that were recorded in this study was searched against only *D. magna* lipids that have been reported previously in the literature [29,31] and against an in-house curated library of *D. magna* lipids [46].

### 3.2. Spatial Lipidome of Daphnia magna

Earlier mass spectrometry studies into the *D. magna* lipidome have primarily focused on the average of all tissue types due to the practical difficulties of separating the tissues in such a small and fragile organism. Employing DESI-MS, we were able to spatially resolve four distinct anatomical regions—appendages, eggs, eye, and the gut—based on the ion images of whole daphnids. Multivariate analysis indicated that each tissue has a unique lipidomic phenotype, specifically that triacylglycerides were predominantly stored in the appendages and eggs, which relates to the maternal energy investment in developing eggs for survival; whilst the phosphatidylcholines were predominantly located in the developing eggs and eye with very little of this lipid class present in the gut or appendages. These results provide a rationale for moving towards measuring toxicity within specific *D. magna* tissues rather than averaging across whole organism homogenates, although the technical challenges of achieving this at scale have yet to be solved. DESI-MS lends itself well to these types of analyses by measuring all the tissue types in a single section with spatial resolution, opening up the possibility of conducting multiple tissue-specific metabolomics studies from a single dataset.

Section 2.3 shows how such an experiment might be conducted, focusing specifically on studying the egg lipidome over time. The lipid changes in a single anatomical region, as detected by DESI-MS, were found to be much more sensitive to the pixel annotation process given the more subtle changes when compared to the analysis of different tissue types in Section 2.2. We found first that subsetting the brood chamber and then annotating the eggs from within these narrowed anatomical regions successfully made the eggs more distinguishable by mitigating the suppression effects of intense pixels in the adult daphnid, which minimized the variation that was caused by annotation errors and improved the data quality.

We detected an expected temporal effect over the seventh instar showing that the abundance of PC(32:00) and other phosphatidylcholines remain steady in the developing egg until 48 h after the sixth brood, before rapidly increasing at 72h as the eggs reached maturity (immediately prior to releasing neonates). The abundance of TG(48:05) and other triacylglycerides in the eggs appeared to steadily decrease over time. Additionally, the ion images indicated that the relative changes in egg lipid abundance were conserved across the storage sites in the whole animal, suggesting that the female daphnids continue to accumulate their own lipids whilst they are simultaneously investing into egg development.

Bisphenol-A has been shown to affect the reproductive fitness and starvation resilience of female offspring of daphnids through perturbations in their lipidome [31]. However, the effect of BPA on the eggs during development has yet to be fully investigated, likely due to the practical challenges of such experiments. DESI-MS enabled us to compare the lipidome of the developing eggs in control female daphnids and those that were exposed to BPA over the seventh adult instar. At the 48 h time point, we found the largest separation between the BPA and the control samples in the PCA scores plot as well as significant variation of the lipidome between the eggs in individual and different daphnid brood chambers which was much less evident upon BPA exposure. This was indicative of lipid perturbation driving the eggs towards a distinct lipidomic phenotype upon BPA exposure 48 h after the sixth brood: the cause of this decrease in phosphatidylcholines and increase in triacylglycerides, both contrary to the general trend across the instar, remains unclear.

There was little evidence of BPA-induced perturbations at the three other time points; in fact, the control and exposed *D. magna* lipidomes appeared to be comparable immediately prior to release of neonates (72 h) based on both single feature and multivariate statistical analyses. This suggests that the lipid perturbation that was evident after 48 h was compensated for to ensure adequate maternal investments. Since previous work proves that BPA has lasting effects on neonates [47], we propose that the specific toxic effects in the developing eggs that were induced 48 h after the sixth brood require further investigation.

## 4. Materials and Methods

### 4.1. Daphnia magna Culturing and Bisphenol-A Exposures

Cultures of *D. magna* were initiated from third brood Bham2 strain neonates and maintained at a density of 20 animals/L in borehole water (collected from the University of Birmingham). The cultures were held at 20 °C with a 16 h:8 h light:dark photocycle and fed daily with *Chlorella vulgaris* (100 µL/animal, 500 cells/µL) as previously reported [48]. Media changes were carried out three times per week during maintenance and daily during chemical exposures.

The single exposure concentration of bisphenol-A (>99%, Merck Life Science Ltd., Gillingham, Kent, UK) in this study was determined by conducting a range finding experiment using 0, 5, 8, 10, 12, and 20 mg/kg with each test vessel containing 500 µL/L of DMSO as a solvent carrier and a total of 8 biological replicates per concentration. Single daphnids in individual 80 mL vessels were exposed immediately after their 6th maternal brood (3–4 weeks old) with samples taken at 8, 24, 48, and 72 h.

For the sample set that was analysed in this work, three biological replicate samples were taken from the control animals and those that were exposed to a single BPA concentration of 5 mg/kg using the same protocol as described for the range finding.

### 4.2. Preparing Tissue Sections for DESI-MS

#### 4.2.1. Preparing Embedding Media

The embedding media was prepared by heating 9% gelatin (Sigma-Aldrich Company Ltd., Poole, Dorset, UK) and 1% carboxymethylcellulose (CMC, Sigma, Poole, Dorset, UK) in 50 mL of HPLC-grade water (VWR, Magna Park, Lutterworth, UK) at 100 °C for 45 min to form a hydrogel. The formulation was then sonicated for 10 mins to remove any air pockets.

#### 4.2.2. Cryosectioning *Daphnia magna*

The live daphnids were individually filtered from the borehole water and embedded in 9% gelatin + 1% CMC (or OCT media during some initial optimization work) until fully submerged. The embedded samples were immediately cooled in LN_2_ vapor until the outer layer had frozen, then submerged in LN_2_ for 1 min. Next, the snap-frozen samples were mounted to a cryostat chuck and left to equilibrate in the cryotome (Bright instruments, OTF 5000/HS-001) at −18 °C. Then, 6 sequential sections of each daphnid were sectioned to 20 µm thickness and thaw-mounted onto pre-cooled frosted microscope glass slides (VWR, SuperFrost, 76 × 26 mm). The sections were stored at −80 °C prior to DESI-MS analysis and H&E staining. The protocols using LN_2_ cooled propane and pentane during optimizations are described in the Appendix A.

#### 4.2.3. Preparing Quality Control Samples

Porcine liver sections for QC purposes were generated and stored using the same method that is described in Section 4.2.2 for the *D. magna* sections. Polyalanine films were generated by pipetting a 3 mg/mL solution of polyalanine (Sigma-Aldrich Company Ltd., Poole, Dorset, UK) that was dissolved in methanol (HPLC grade, VWR, Magna Park, Lutterworth, UK) onto frosted glass slides and heating to 50 °C until the methanol evaporated. The films were stored under ambient conditions prior to MS analysis and remained viable for two months.

### 4.3. Acquiring DESI-MS Data

#### 4.3.1. Preparing the Desorption Electrospray Ionization—Mass Spectrometer

Molecular analysis of the *D. magna* sections was carried out with a Prosolia 2D DESI source and Xevo G2-XS QToF mass spectrometer (Waters, Wilmslow, UK). Prior to data acquisition, the performance of the MS system was evaluated and optimized following the manufacturer’s recommendations using the LockSpray ESI source (Waters, Wilmslow, UK). The DESI source was then attached and the movement of the xy-plate was tested by analyzing rhodamine 6G dye in red Sharpie^®^ ink [37]. Next the DESI geometric parameters were optimized by analyzing a porcine liver section. Specifically, the DESI source to sample distance and source to sample angle were adjusted to ensure the trajectory of analytes in 2o DESI droplets were directed towards the mass spectrometer orifice.

#### 4.3.2. Analyzing *Daphnia magna* Study Samples

Prior to DESI-MS analysis of the *D. magna* tissue sections: (i) a marker pen was used to outline the sample area on the underneath of the sample slide, (ii) an optical image of the slide was generated, (iii) this was imported into HDImaging v1.5 (Waters, Wilmslow, UK) where the analysis ROI was defined using the markers, and (iv) the DESI-MS acquisition parameters were set, listed below. The experiment files were then imported into MassLynx v4.2 (Waters, Wilmslow, UK), the analysis slide that was positioned on the DESI sample holder and data acquisition was carried out in positive ion, sensitivity mode across the 50–1000 *m*/*z* range using a pixel size of 35 um^2^ that was scanned at 1 pixel/s. The DESI solvent (95% MeOH) flowed at a rate of 2 µL/min through the heated DESI capillary (prototype from Waters) that was held at 0.7 kV and 400 °C and was pneumatically directed towards the ion block (150 °C) by 1 Bar N_2_ (g).

#### 4.3.3. Quality Control Strategy

During data acquisition, rhodamine 6G dye (red ink) was added to each sample slide and analysed before the biological sample to monitor the *m*/*z* drift over time. In instances where the *m*/*z* error exceeded 25 ppm, a polyalanine film was analysed to determine the *m*/*z* drift across the entire mass range and, where appropriate, was corrected to the detectable polyalanine fragment ions (Appendix A). The signal stability was monitored before each sample by analyzing the lipid profile in porcine liver sections. When the signal had significantly decayed (intensity < 1 × 10^5^), the DESI parameters were re-optimized such that the porcine liver lipid profile had an intensity of 1 × 10^5^. In this scenario, data from the previous sample was excluded from the study and a sequential section was analysed. At the start and end of the data acquisition, a blank DESI slide was analysed to generate data for the removal of background ions.

### 4.4. Data Processing and Statistical Analysis of Daphnia magna DESI-MS Data

The raw data were converted to imzML file format [49] using HDImaging which was then read into R (v4.0.0) using the open source package Cardinal (v2.6.0). Processing the MSI dataset was then primarily conducted in Cardinal initially selecting ROIs around *D. magna* tissues from each tissue section using the ellipse tool on the ion images (Appendix A).

The pre-processing steps were carried out on the resulting data as follows: the features with an SNR < 3 in individual spectra were removed (the noise level was calculated using the median absolute deviation); the remaining features were aligned within the individual tissue sections using a 25 ppm tolerance; the features that were present in less than 5% of pixels within a tissue section were removed; the intensities within each spectra were normalized to the square root of the mean squared intensities (RMS); the normalized features across all the tissue sections were aligned using a 35 ppm tolerance; and the features from the median spectra of all the *D. magna* pixels with <10× intensity in the DESI blank data were removed. The combined dataset was then subset, further processed, and analyzed to address specific objectives.

For the results Section 2.2, the pixels pertaining to the eggs, eye, gut, and appendages were extracted across the entire sample set (as for extracting *D. magna* tissue pixels). A total of 200 pixels from each tissue type were sampled to create a balanced dataset and prevent a single tissue dominating the multivariate analyses. Then, a 50% pixel filter and missing value filter was applied and pixels that were lying outside a 95% confidence ellipse from the PCA of pixels from each tissue type (the independent variable) were removed. For multivariate analysis, the matrices underwent missing value imputation (k-nearest neighbours, k = 5) and pareto scaling.

For Section 2.3 and Section 2.4, the pixels relating to eggs in the *D. magna* sections were subset after initially setting an ROI around the brood chamber for improved contrast and annotation quality. For Section 2.3, the control samples from the 8, 24, 48, and 72 h time points were processed, and for Section 2.4, the control and exposed samples from each time point were processed separately. Further processing was analogous to Section 2.2 where the independent variables were the 8 to 72 h time points (Section 2.3) and the treatment (Section 2.4), respectively.

The MS1 data were searched against the reported and an in-house curated library of *D. magna* lipids [46] within a 25 ppm tolerance.

Data from the processing steps that were outlined above were analyzed in three parts: (i) the ion images where the distribution of individual lipids was mapped across the individual daphnids by plotting the intensity at each given pixel as a scaled color from the vidris color map; (ii) unsupervised multivariate analysis which consisted of a PCA, and (iii) the individual feature analysis where the intensity distribution plots were generated as a function of the anatomical region, egg development stage, and treatment.

For the analyses in results Section 2.2, each anatomical region comprised of the intensities of given features in all the pixels across all the tissue sections such that the number of individual data points (n) equalled the number of pixels in each group (tissue type).

For the analyses in Section 2.3 and Section 2.4, a median intensity of each feature across all the egg pixels within the individual tissue sections was taken forward. Therefore, in these analyses, the number of individual data points (n) equalled the number of biological replicates. A Wilcoxon signed-rank test was performed to compare the lipid abundances between the control and the BPA-exposed tissue section in Section 2.4, using Benjamini and Hochberg’s false discovery rate (FDR) correction to account for multiple testing [50].

## 5. Conclusions

The best practices for histology, DESI-MS, and data processing were adapted to successfully generate the first spatially-resolved mass spectrometry measurements of *D. magna*. The approach was able to spatially characterize the baseline lipidome of *D. magna*, identifying differences between the distinct anatomical regions (eggs, eye, gut, and appendages). The baseline lipidome of the developing eggs were further investigated in female daphnids across their seventh instar, indicating an increasing abundance of phosphatidylcholines and a decreasing abundance of triacylglycerides from maternal investment. We then investigated how this developmental process was perturbed upon BPA exposure, specifically 48 h after the sixth brood, characterized by increased triacylglyceride and decreased phosphatidylcholine abundance. Overall, we believe this work demonstrates the capabilities of DESI-MS (and MSI techniques as a whole) as a tool to visualize and investigate the lipidomic differences between distinct tissue types in *Daphnia magna,* offering improved biochemical understanding of toxicity, for instance to identify sites of toxicity.

## Figures and Tables

**Figure 1 metabolites-12-00033-f001:**
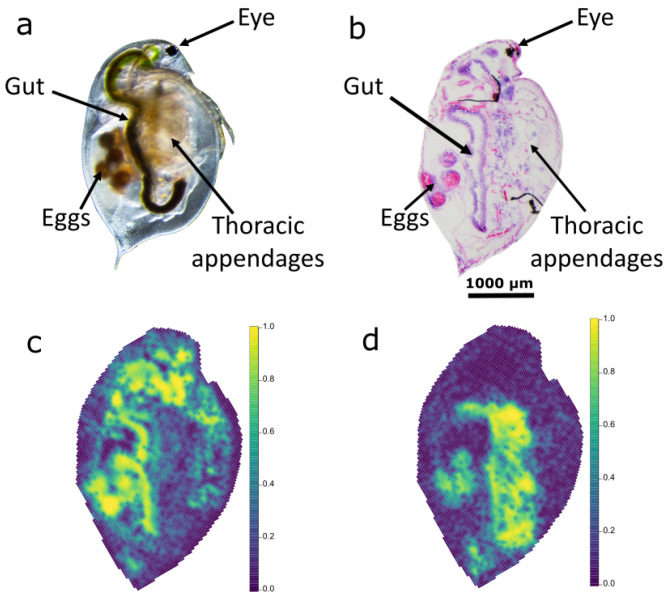
Images of the model organism *Daphnia magna*; (**a**) labelled microscope image, (**b**) labelled H&E-stained cryosection that was generated using our optimized tissue preparation workflow, (**c**,**d**) ion images showing the distribution of phosphatidylcholine (*m*/*z* = 756.5521) and triacylglyceride (*m*/*z* = 835.6229), respectively. The color bars show the relative intensity of the lipid in the ion images.

**Figure 2 metabolites-12-00033-f002:**
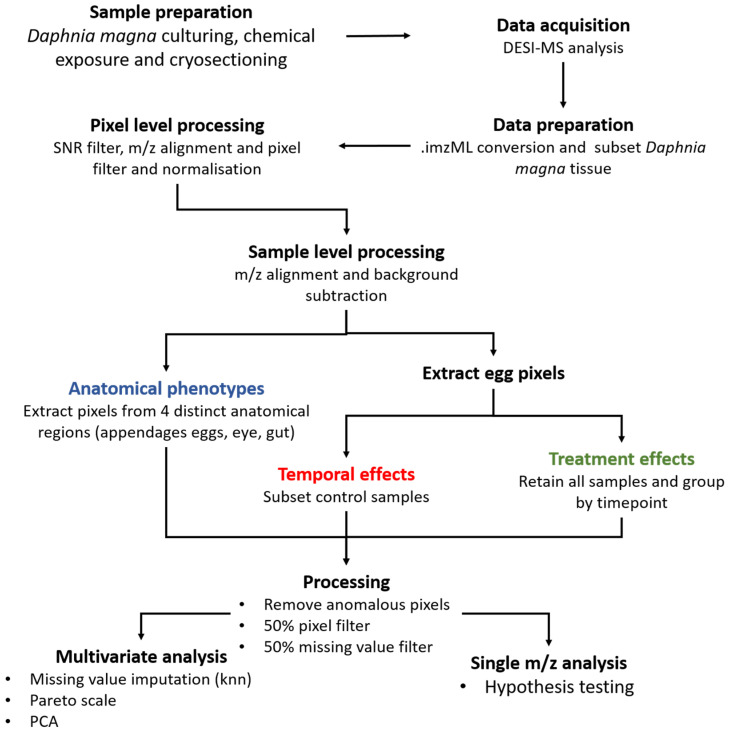
DESI mass spectrometry imaging workflow from culturing to statistical analysis for spatial lipidomic studies of *Daphnia magna*.

**Figure 3 metabolites-12-00033-f003:**
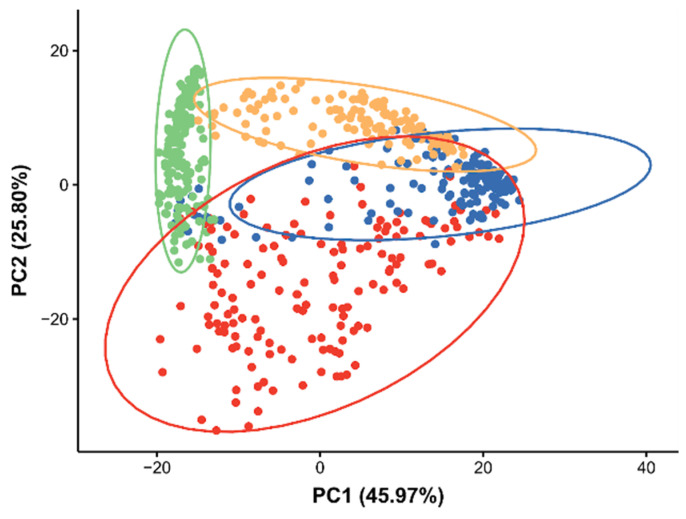
PCA scores plot from analysis of the DESI-MS dataset of four distinct tissues types in *Daphnia magna*, illustrating the differences in the baseline lipidomes of the appendages (blue), eggs (yellow), eye (green), and the gut (red). Each point relates to a single pixel from a DESI-MS ion image of *D. magna*.

**Figure 4 metabolites-12-00033-f004:**
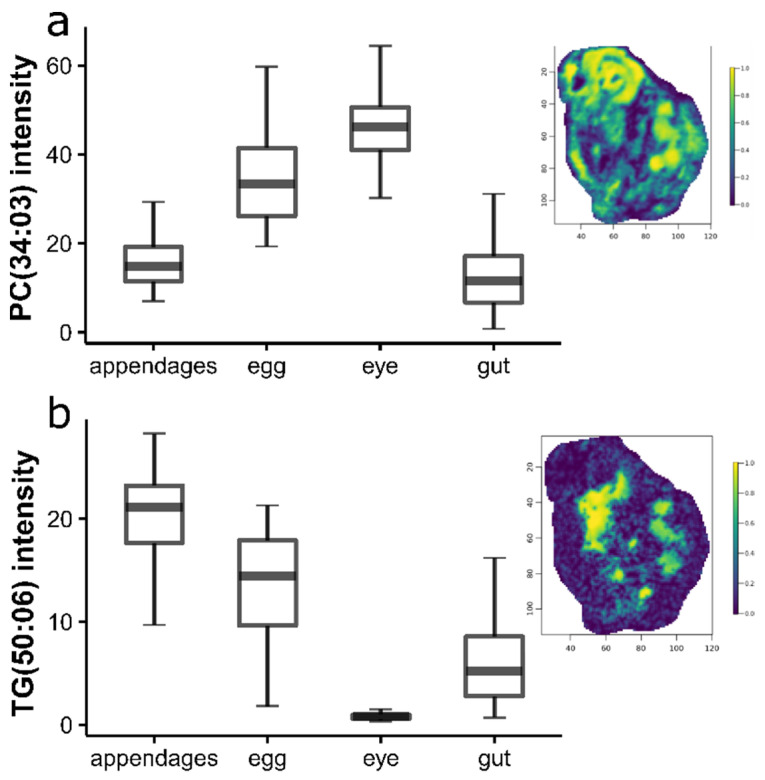
Boxplots showing the differences in the abundance of a representative phosphatidylcholine and triacylglyceride; (**a**) PC(34:03) at *m*/*z* = 778.5347, and (**b**) TG(50:06) at *m*/*z* 861.6503, between *Daphnia magna* appendages (blue), eggs (yellow), eye (green), and gut (red) following DESI-MS analysis. The error bars represent a 95% confidence interval. The representative ion images are overlaid on each plot to visualize the distribution of each lipid across the whole animal. The colour scale indicates the relative abundance of each lipid.

**Figure 5 metabolites-12-00033-f005:**
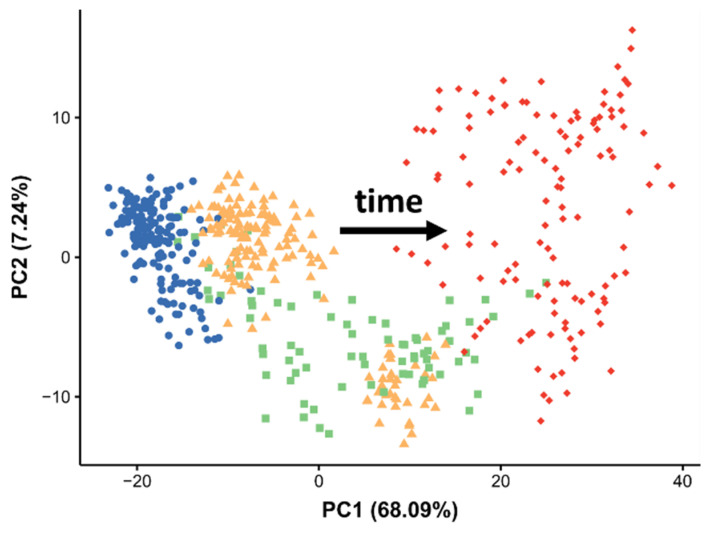
PCA scores plot from analysis of the DESI-MS dataset of unexposed *Daphnia magna* eggs across the seventh adult instar—specifically 8 (blue circles), 24 (yellow triangles), 48 (green squares), and 72 h (red diamonds) after the sixth brood. Each point relates to a single pixel from a DESI-MS ion image of *D. magna* egg.

**Figure 6 metabolites-12-00033-f006:**
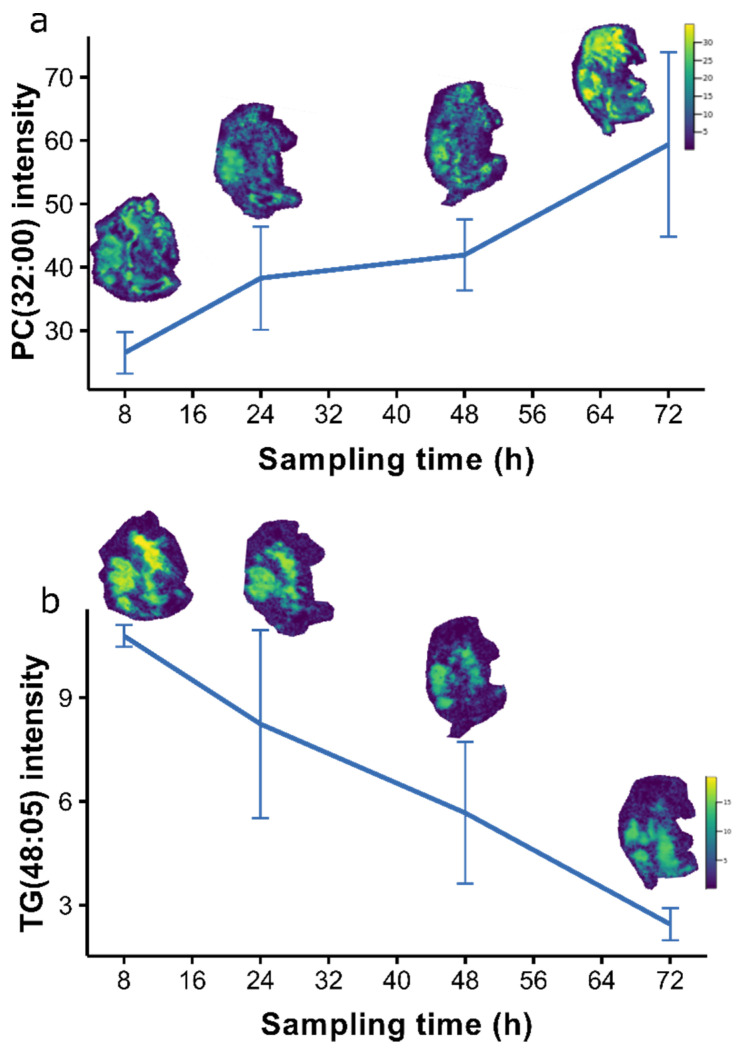
Line graphs showing the differences in the abundance of a representative phosphatidylcholine and triacylglyceride; (**a**) PC(32:00) at *m*/*z* = 756.5521, and (**b**) TG(48:05) at *m*/*z* 835.6229, in *Daphnia magna* eggs across the seventh adult instar—specifically 8, 24, 48, and 72 h after the sixth brood. The error bars represent a 95% confidence interval. The ion images are overlaid on each plot to visualize the distribution of each lipid across the whole animal over time. The colour scale indicates the relative abundance of each lipid.

**Figure 7 metabolites-12-00033-f007:**
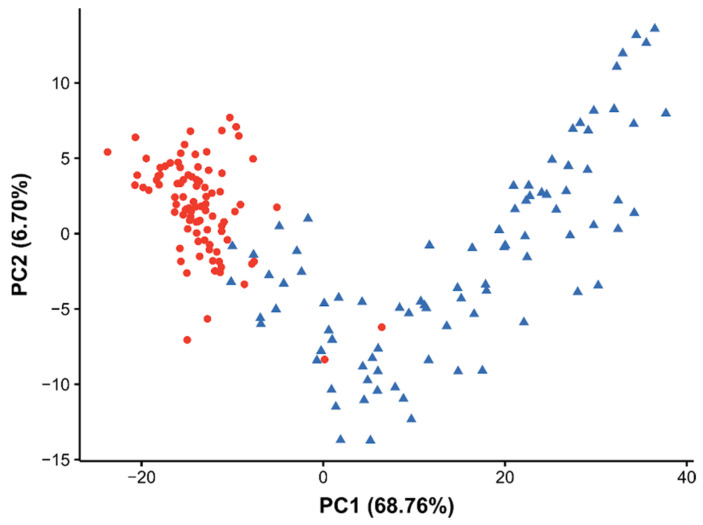
The PCA scores plot from analysis of the DESI-MS dataset of the control *Daphnia magna* eggs (blue triangles) relative to those that were exposed to 5 mg/kg bisphenol-A for 48 h (red circles). The exposure period started immediately after the sixth adult brood. Each point relates to a single pixel from a DESI-MS ion image of a *D. magna* egg.

**Figure 8 metabolites-12-00033-f008:**
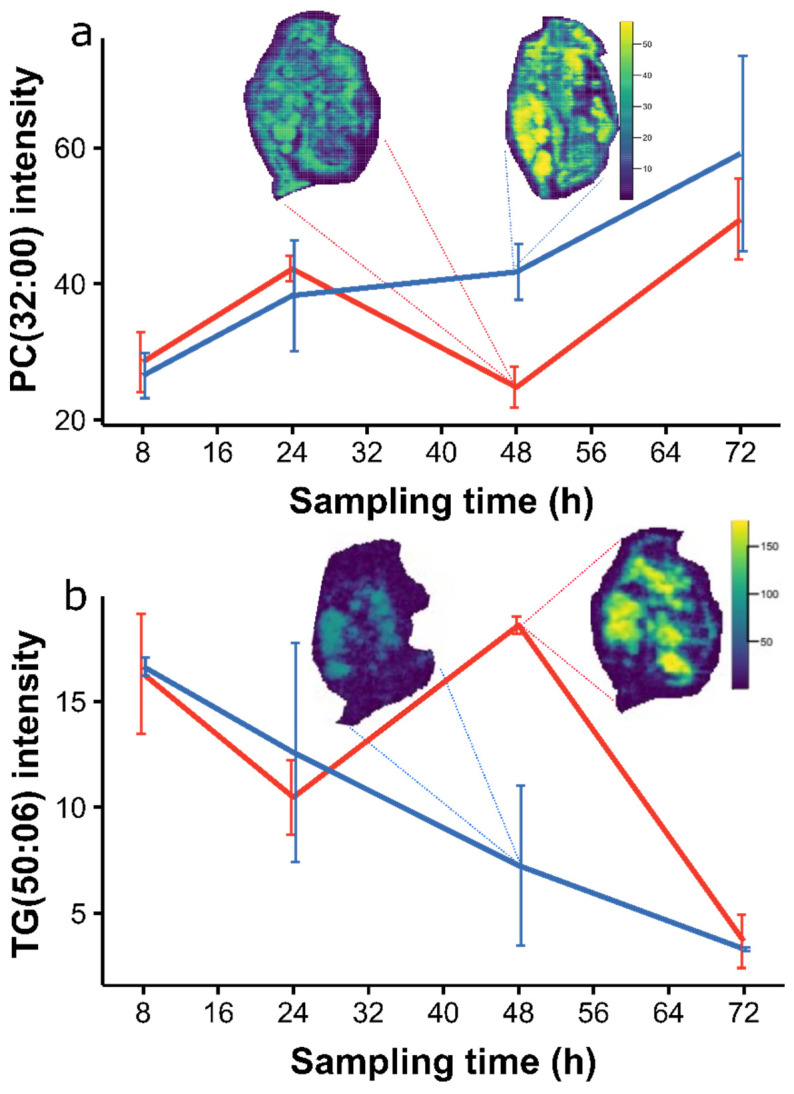
Line graphs showing the differences in abundance of a representative phosphatidylcholine and triacylglyceride; (**a**) PC(32:00) at *m*/*z* = 756.5521, and (**b**) TG(50:06) at *m*/*z* 861.6503, in control (blue) and bisphenol-A exposed (red) *Daphnia magna* eggs across the seventh adult instar—specifically 8, 24, 48, and 72 h after the sixth brood. The error bars represent a 95% confidence interval. The ion images are overlaid on each plot to visualize the distribution of each lipid across the whole animal as a function of BPA exposure 48 h into the instar. The colour scale indicates the relative abundance of each lipid.

## Data Availability

The data presented in this study are openly available in Zenodo at https://doi.org/10.5281/zenodo.5680037 (12 November 2021).

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
