# Peer review of "Spatially Mapping the Baseline and Bisphenol-A Exposed Daphnia magna Lipidome Using Desorption Electrospray Ionization—Mass Spectrometry"

_metabolites, 2022, doi:10.3390/metabo12010033_

Round 1

Reviewer 1 Report

The manuscript reports spatial and temporal metabolome changes of D. magna using DESI-mass spectrometry. It is quite interesting approach that many readers of metabolites journal would be interested in.  However, the current manuscript has several unclear part to be revised thoroughly. 

specific comments

  1. line 169: I think that metabolomics is not equal to lipidomics. Among various metabolites, there are lipid metabolites. The authors began the manuscript with metabolomics and suddenly changed their topic to lipidomics.  I could not find any explanation how they recognized lipids among general metabolites except that they mentioned "Lipid profile between m/z 700 and 900" in Fig. S3. Therefore, I assume that they identified their peaks between m/z 700 and 900 as lipids but there were no specific explanation for this matter.  I highly recommend the authors should include explanations on how they identify lipid substances.
  2. line 233: Fig S5 shows a loading plots for Fig. 3 without indicating which are phosphatidylcholine (PC) and triacylglycerides(TG). Since they did not indicate PC and TC in Fig S5, it is impossible to confirm their argument that PC and TG were important for separation of tissue types. 
  3. line 241: I would like to know what 34:03 and 50:06 are. I guess that these are retention time for PC and TG.  Please explicitly explain these numbers. 
  4. line 270: The authors argue that PC and TG are dominant for PC1 and again both PC and TG were not indicated in FIG S6.
  5. line 275: Therefore, it is hard to follow the author's argument of increasing PC and decreasing TG over time. 
  6. Figure 6: Figure 6 is duplicated in Figure 8. Figure 8 shows PC and TG changes in both control and BPA treatment while Figure 6 shows PC and TG in control only.  Therefore, I recommend the Figure 6 could be removed for simplicity. 
  7. Figure 7: Figure 7 is one of 4 PCA from S8. The authors would better to remove Figure 7 or present Fig S8 instead of Figure 7.
  8. line 292: The authors argues that EC50 was 14 ppm based on Figure S7. However, in Figure S7, EC50 appears to be 1.4 ppm rather than 14 ppm. Please present a figure more focusing on 1 to 20 ppm interval so that readers can confirm EC50. 

Author Response

We thank reviewer 1 for their comments and have addressed them as follows:

  1. line 169: I think that metabolomics is not equal to lipidomics. Among various metabolites, there are lipid metabolites. The authors began the manuscript with metabolomics and suddenly changed their topic to lipidomics.  I could not find any explanation how they recognized lipids among general metabolites except that they mentioned "Lipid profile between m/z 700 and 900" in Fig. S3. Therefore, I assume that they identified their peaks between m/z 700 and 900 as lipids but there were no specific explanation for this matter.  I highly recommend the authors should include explanations on how they identify lipid substances.
    1. We agree that metabolomics is not equal to lipidomics, however the workflows we have used for lipidomics are derived from metabolomics. With edits on lines 168 and 184 we feel we now make this distinction clear. The text (formerly on) line 169 described a lipid profile from porcine liver used to optimise the DESI-MS system and is a common method to do so - a reference has been added.
  2. line 233: Fig S5 shows a loading plots for Fig. 3 without indicating which are phosphatidylcholine (PC) and triacylglycerides(TG). Since they did not indicate PC and TC in Fig S5, it is impossible to confirm their argument that PC and TG were important for separation of tissue types. 
    1. We agree this was not clear and have added the identified lipid names to the loadings plot.
  3. line 241: I would like to know what 34:03 and 50:06 are. I guess that these are retention time for PC and TG.  Please explicitly explain these numbers. 
    1. We have added explicit detail about the standard naming format for lipids at its first use on line 257.
  4. line 270: The authors argue that PC and TG are dominant for PC1 and again both PC and TG were not indicated in FIG S6.
    1. We agree this was not clear and have added the identified lipid names to the loadings plot.
  5. line 275: Therefore, it is hard to follow the author's argument of increasing PC and decreasing TG over time. 
    1. Line 287 states that the argument is based on Figure 6 (not Figure S6) which shows the intensity decrease and increase in the form of a line graph for the PC and TG lipids respectively. We therefore don’t feel we need to make an edit for this comment.
  6. Figure 6: Figure 6 is duplicated in Figure 8. Figure 8 shows PC and TG changes in both control and BPA treatment while Figure 6 shows PC and TG in control only.  Therefore, I recommend the Figure 6 could be removed for simplicity.
    1. We thank the reviewer for highlighting this point because it led to us spotting an error in Figure 6, specifically we have wrongly labelled the y-axes (both the axis labels and Figure caption have been corrected). In fact, the data in Figures 6 and 8 are not identical as Figure 6 corresponds to TG(48:05) and Figure 8 corresponds to a different triacylglyceride (TG(50:06)). We have corrected Figure 6 to ensure this is more clear.
  7. Figure 7: Figure 7 is one of 4 PCA from S8. The authors would better to remove Figure 7 or present Fig S8 instead of Figure 7.
    1. While we appreciate the reviewers comment, which arises because Figure 7 is the same as Figure S8c, we believe Figure 7 is the optimal one for inclusion in the main paper to make the paper easier to follow (since the other PCA scores plots are less relevant for the text in the main paper). We could delete Figure S8c from Figure S8, but then we believe Figure S8 loses some value - which is supported by the comments from revierer 3 who claims it is a  “very nice figure”.
  8. line 292: The authors argues that EC50 was 14 ppm based on Figure S7. However, in Figure S7, EC50 appears to be 1.4 ppm rather than 14 ppm. Please present a figure more focusing on 1 to 20 ppm interval so that readers can confirm EC50. 
    1. We suspect, based on the reviewers comment, that they may not have realised the data is presented on a log scale. However, as requested, we have added a further image zoomed in on the x-axis so that our results are more clear.

Reviewer 2 Report

While untargeted lipidomics toxicity studies have been performed for D. magna--primarily by LC-MS--this is the first report to use mass spectrometry imaging (MSI) technology to obtain spatial distribution of lipid species across organs of D. magna. Thus, the authors were able to examine distributions of lipids to different anatomic regions of the organism by MSI. In this study, DESI was used as the ionization source for MSI, which is highly appropriate. There are a number of novel features regarding this study. First, using principle component analysis, the lipid distribution amongst anatomic features could be differentiated, leading to distinct lipid profiles for eggs, eyes, appendages, and guts in D. magna. Another novel feature was tracking lipidomic changes across the various stages of the egg life cycle, and then repeating the study except using exposure to bisphenol A to examine the impact it has on lipid distribution. PCs and TAGs were the major classes of lipids perturbed by bisphenol A exposure which is consistent with prior LC-MS whole homogenate studies. The authors very carefully developed an optimized DESI-MS workflow and were able to obtain whole body images of these organisms. This is a marvelous application of MSI generally, and DESI in particular, to studying toxicological effects in crustaceans in different anatomic tissues. This paper is quite appropriate for publication in Metabolites. There are only a few minor comments for the authors to consider below.

  1. line 114. "distinguish" should be "distinguished" in this instance; the authors attribute a pixel size of 35 square micrometers, but they do not state how they obtained this determination. Was it done after a microscopic investigation to determine the size of an ablation crater on the tissue surface?
  2. line 369. The authors make note of the potential to use orthogonal MSI approaches like MALDI. Indeed, a complementary DESI and MALDI MSI study on rat brain tissue appeared in this journal earlier in the year: Metabolites 2021, 11, 253. The authors may find that paper of interest.

Author Response

We thank reviewer 2 for their comments and have addressed them as follows:

  1. line 114. "distinguish" should be "distinguished" in this instance; the authors attribute a pixel size of 35 square micrometers, but they do not state how they obtained this determination. Was it done after a microscopic investigation to determine the size of an ablation crater on the tissue surface?
    1. Distinguish changed as suggested. Pixel size was not determined, instead set in the instrument software based on assumptions of the capabilities of the DESI system. We therefore accept that we cannot explicitly claim the pixel size was exactly 35 um2. We have changed the text on line 119 to state that this “pixel size was set to 35 um2 in the instrument software” to reflect this.
  2. line 369. The authors make note of the potential to use orthogonal MSI approaches like MALDI. Indeed, a complementary DESI and MALDI MSI study on rat brain tissue appeared in this journal earlier in the year: Metabolites 2021, 11, 253. The authors may find that paper of interest.
    1. We agree this claim should be supported by reference to prior work and have included the paper that the reviewer highlighted.

Reviewer 3 Report

In their manuscript the authors describe a novel approach for desorption electrospray ionisation - mass spectrometry (DESI-MS) applied to the lipidome analysis of daphniids. Their approach allows the spatial mapping in sections. Firstly, the manuscript is technically sound and written in a very nice professional language style and formatting. The study presents something quite novel and important to the field of ecotoxicology with a research focus in daphniids, and is worth reading and considering for publication.

Some suggestions regarding improvements of the text

Abstract

Change the first sentence to:

Untargeted lipidomics has previously been applied to the study of daphniids and the discovery of biomarkers indicative of toxicity.

Change the sentence to:

Here we applied mass spectrometry imaging of Daphnia magna to combine untargeted lipidomics with spatial resolution to map molecular perturbations to defined anatomical regions.

Change the sentence to:

A desorption electrospray ionisation - mass spectrometry (DESI-MS) method was optimised and applied to tissue sections of daphniids exposed to bisphenol-A (BPA) against unexposed controls, generating an untargeted mass spectrum at each pixel (35 µm2/pixel) within each section.

Introduction

Line 42: (AOP) change to (AOPs)

Line 43: will focus on detecting molecular change to will focus on the detection of molecular

Line 45: comprises early response molecular changes to comprises of early response molecular

Line 59: have required the homogenisation change to require the homogenisation

Materials and Methods

Line 488 Daphnia magna should be in italics

Results

Figure 5 is a very good figure to show the effect on the BPA treatment time. However, if I understood correctly, the figure only shows the exposed group. What about the controls at the same time points? How do the controls behave at the same time points? Why did the authors not include them in this figure? I understand that the different time points between control and BPA are in figure S8, so I am covered if the authors decide to leave the figure 5 for clarity as it is in the end. It would be nice, however, if all the controls of different time points group in one region in the PCA plot. Perhaps this is something to consider if it is separating well. Perhaps the authors can consider this suggestion in figure S8 before they separate the panels per time point.

Supplementary

Figure S1: What is the y axis? It is mentioned as mean. Mean of what?

Figure S5: This is a nice figure, however, I would like some clarifications. The authors mention “PCA loadings plot from the DESI-MS analysis of 4 distinct tissues types in Daphnia magna sections - appendages, egg, eye and gut”. This PCA is very nice and PC1 and PC2 together have approximately 71% which is a good score for the variance. I see red and black dots with numbers for the red. Are the red values i.e. the m/z peaks they analysed as loadings? Please be more detailed in the figure legend. What are the numbers? What are the colours? Is something corresponding to the tissue type here? Why there are no numbers in the black points? What do they illustrate? A reader who is not in the field may not understand the message of this figure and the figure legend should be concise and self-explanatory.

Figure S6: This is a nice figure too, but clarifications might be necessary similar to figure S5. Please explain where do we see the time effect here?

Figure S8: Again very nice figure. There are 4 sub figures. Please label which is 8, 24, 48 and 72 hours. There is a good separation of treatments in the tissue of eggs and this is very good data. Please check comment on figure 5.  

Figure S9: Some clarifications may be necessary. The authors state “Each pixel is coloured based on the treatment group and replicate it was taken from - highlighting the heterogeneity between eggs from the same daphnid and different replicates.” I see triangles (dark red and blue). What group is this? Is it only 2 different (control or BPA?) replicates, hence the different colours? I see circles (green, purple, orange). Which group is this (control or BPA?) and what do the three colours mean? Is it different animals?

Figure S10: Please explain more as per previous comments for Figure S5.

Some questions for the authors to consider

Clarification on rhodamine: I understand that rhodamine 6G used as a stability standard and as such was this added in the sections or not? At what concentration? In line 483 rhodamine B is mentioned. Which one was it? 6G or B?

Clarification on exposures: Lines 443 to 451. The authors describe how they chose the concentration range and that they used animals after their 6th maternal brood. My first question is how old are these animals? Another question is why did the authors choose animals after 6th brood? I assume these animals are probably “emptying” from eggs as daphniids make a peak brood in the 3rd and then they start to reduce typically. Also could the authors specify how many animals per 80 ml volume they exposed? Line 446 states “Daphniids in individual 80 ml vessels…”. Was it one animal in 80 ml or more? Please specify.

Classification on identifying m/z to specific lipids annotated: The authors cite their review paper on annotation of metabolomes in line 383. They also cite two research articles in the field. Taking into account that submission of mass peaks is a useful practice to follow in the field of metabolomics, I would request the authors to provide for one or two of their experiments at least the final peak list of m/z with lipid annotation along with the method of annotation (i.e. which library or reference they used) and the counts (m/z, annotation, method identified, counts) that they used for their statistical analysis. Such a file as a csv or Excel should be very small in size too. Perhaps two good datasets would be sufficient (such as the ones of figures 5 and 7) to aid the reader to understand the paper better. 

Author Response

We thank reviewer 3 for their comments and have addressed them as follows:

Abstract

Change the first sentence to:

Untargeted lipidomics has previously been applied to the study of daphniids and the discovery of biomarkers indicative of toxicity.

Change the sentence to:

Here we applied mass spectrometry imaging of Daphnia magna to combine untargeted lipidomics with spatial resolution to map molecular perturbations to defined anatomical regions.

Change the sentence to:

A desorption electrospray ionisation - mass spectrometry (DESI-MS) method was optimised and applied to tissue sections of daphniids exposed to bisphenol-A (BPA) against unexposed controls, generating an untargeted mass spectrum at each pixel (35 µm2/pixel) within each section.

 Introduction

Line 42: (AOP) change to (AOPs)

Line 43: will focus on detecting molecular change to will focus on the detection of molecular

Line 45: comprises early response molecular changes to comprises of early response molecular

Line 59: have required the homogenisation change to require the homogenisation

 Materials and Methods

Line 488 Daphnia magna should be in italics

  • All of the edits to the ‘abstract’, ‘introduction’ and ‘Materials and Methods’ have been implemented as suggested.

 Results

Figure 5 is a very good figure to show the effect on the BPA treatment time. However, if I understood correctly, the figure only shows the exposed group. What about the controls at the same time points? How do the controls behave at the same time points? Why did the authors not include them in this figure? I understand that the different time points between control and BPA are in figure S8, so I am covered if the authors decide to leave the figure 5 for clarity as it is in the end. It would be nice, however, if all the controls of different time points group in one region in the PCA plot. Perhaps this is something to consider if it is separating well. Perhaps the authors can consider this suggestion in figure S8 before they separate the panels per time point.

  • Figure 5 shows the control (unexposed) samples only. We have edited the text on line 269 and in the caption for Figure 5 to make this clearer. We believe this covers the other points too.

Supplementary

Figure S1: What is the y axis? It is mentioned as mean. Mean of what?

  • The y-axis has been edited to read “Section quality score”

Figure S5: This is a nice figure, however, I would like some clarifications. The authors mention “PCA loadings plot from the DESI-MS analysis of 4 distinct tissues types in Daphnia magna sections - appendages, egg, eye and gut”. This PCA is very nice and PC1 and PC2 together have approximately 71% which is a good score for the variance. I see red and black dots with numbers for the red. Are the red values i.e. the m/z peaks they analysed as loadings? Please be more detailed in the figure legend. What are the numbers? What are the colours? Is something corresponding to the tissue type here? Why there are no numbers in the black points? What do they illustrate? A reader who is not in the field may not understand the message of this figure and the figure legend should be concise and self-explanatory.

  • The caption has now been updated with the following text added to clarify the plot; “Red points indicate pixels outside a 95% ellipse presumed to be important for distinguishing tissue types. Labels represent the putatively annotated lipid where possible or m/z value of each feature.”

Figure S6: This is a nice figure too, but clarifications might be necessary similar to figure S5. Please explain where do we see the time effect here?

  • We did not include the time axis on the loadings since it is less conventional to do so. However, based on information from the scores plot, we have superimposed the arrow to aid visualisation and explicitly outlined how this was derived (i.e., from the corresponding scores plot) in the figure caption.

Figure S8: Again very nice figure. There are 4 sub figures. Please label which is 8, 24, 48 and 72 hours. There is a good separation of treatments in the tissue of eggs and this is very good data. Please check comment on figure 5.  

  • We thank the reviewer for pointing out this oversight and we have labelled the plot as suggested.

Figure S9: Some clarifications may be necessary. The authors state “Each pixel is coloured based on the treatment group and replicate it was taken from - highlighting the heterogeneity between eggs from the same daphnid and different replicates.” I see triangles (dark red and blue). What group is this? Is it only 2 different (control or BPA?) replicates, hence the different colours? I see circles (green, purple, orange). Which group is this (control or BPA?) and what do the three colours mean? Is it different animals?

  • Figure S9 shows the same PCA as Figure 7, however rather than colouring by treatment (as in Figure 7) the colours represent different animals. The aim of the plot is to highlight egg heterogeneity so we believe only this information is necessary. The plot has been updated such that all points are the same shape and the caption has been updated to make this rationale of the plot clearer.

Figure S10: Please explain more as per previous comments for Figure S5.

  • Updated as for the other loadings plots.

Some questions for the authors to consider

Clarification on rhodamine: I understand that rhodamine 6G used as a stability standard and as such was this added in the sections or not? At what concentration? In line 483 rhodamine B is mentioned. Which one was it? 6G or B?

  • Rhodamine B is a mistake and has been edited to 6G. We thank the reviewer for pointing out the mistake. It was added to the DESI slides with a red sharpie to monitor m/z drift (not intensity) so we didn’t think the concentration was necessary and therefore did not measure it. In the text a reference is present immediately after where the procedure is described in more detail. However we have clarified the procedure on line 509.

Clarification on exposures: Lines 443 to 451. The authors describe how they chose the concentration range and that they used animals after their 6th maternal brood. My first question is how old are these animals? Another question is why did the authors choose animals after 6th brood? I assume these animals are probably “emptying” from eggs as daphniids make a peak brood in the 3rd and then they start to reduce typically. Also could the authors specify how many animals per 80 ml volume they exposed? Line 446 states “Daphniids in individual 80 ml vessels…”. Was it one animal in 80 ml or more? Please specify.

  • Lines 122 and 308 already outline the rationale for animals to be selected after their 6th brood.
    • “we chose to study adult D. magna (3–4-week-old) to maximise the tissue size.”
    • “to ensure the animals were of sufficient size for DESI-MS”
  • The text has been edited to clarify the number of daphnids per vessel and age of daphnids (line 471-472).

Classification on identifying m/z to specific lipids annotated: The authors cite their review paper on annotation of metabolomes in line 383. They also cite two research articles in the field. Taking into account that submission of mass peaks is a useful practice to follow in the field of metabolomics, I would request the authors to provide for one or two of their experiments at least the final peak list of m/z with lipid annotation along with the method of annotation (i.e. which library or reference they used) and the counts (m/z, annotation, method identified, counts) that they used for their statistical analysis. Such a file as a csv or Excel should be very small in size too. Perhaps two good datasets would be sufficient (such as the ones of figures 5 and 7) to aid the reader to understand the paper better. 

  • We fully accept this suggestion, however we also would like to point out that our annotations are “putative, based on experimental MS1 data” (line 405). Therefore we feel that such a table may lead to a reader misinterpreting the level of confidence in these annotations. However, to address the concern, for each loadings plot in the supplementary we have included the putative IDs (lipid class, fatty acid chain length and double bonds) for the interesting features, i.e., those outside the 95% ellipse.

Reviewer 4 Report

The article would be of value to the readership of Metabolites as it delivers an insightful approach to map the bisphenol-A exposed lipidome of D. magna. However, the authors need to make a few revisions to improve the overall quality of the manuscript. First of all, the authors should make clearer to the readers the advantages of mapping the lipid changes due to bisphenol A exposition.

In the introduction the authors should improve the literature overview by including other studies reporting the  detection and visualization of contaminants as well as chemical changes due to exposition to pollutants and drugs by DESI MS imaging. From a quick search on pubmed the following papers were retrieved:

1)Pirro V et al Lipid dynamics in zebrafish embryonic development observed by DESI-MS imaging and nanoelectrospray-MS. Mol Biosyst. 2016 Jun;12(7):2069-79. doi: 10.1039/c6mb00168h.

2) Perez CJ, Monitoring Toxic Ionic Liquids in Zebrafish (Danio rerio) with Desorption Electrospray Ionization Mass Spectrometry Imaging (DESI-MSI). J Am Soc Mass Spectrom. 2017 Jun;28(6):1136-1148. doi: 10.1007/s13361-016-1515-9.

3) Zhao X, Huang X, Zhang X, Shi J, Jia X, Zhu K, Shao B. Distribution visualization of the chlorinated disinfection byproduct of diazepam in zebrafish with desorption electrospray ionization mass spectrometry imaging.Talanta. 2022 Jan 15;237:122919. doi: 10.1016/j.talanta.2021.122919.

Therefore, the authors should make an effort to improve their literature review.

Moreover, the authors made a huge effort in explaining the evolvement  of D. magnain biological testing, but almost nothing was written about the hazard of bisphenol A (they wrote a couple of sentences at lines 416-417). They should emphasize this point in the introduction, too.

Line 444. They should also explain in the introduction why they choose to expose the  D. magna to such high concentrations of bisphenol A(5, 8, 10, 12, 20 ppm).  Are they reasonable concentrations easily encountered in real life? I do not think so. Are there any protocols suggesting such high concentrations for testing biological changes due to hazards expositions? Please provide a suitable explanation.

Line 444 Please change ppm to mg/kg.

I wonder if the authors were able to detect/map bisphenol A in the samples.

Line 360 ESI LC MS does not exist. The authors meant LC-ESI-MS, I guess.

Author Response

We thank reviewer 4 for their comments and have addressed them as follows:

In the introduction the authors should improve the literature overview by including other studies reporting the  detection and visualization of contaminants as well as chemical changes due to exposition to pollutants and drugs by DESI MS imaging. From a quick search on pubmed the following papers were retrieved:

1)Pirro V et al Lipid dynamics in zebrafish embryonic development observed by DESI-MS imaging and nanoelectrospray-MS. Mol Biosyst. 2016 Jun;12(7):2069-79. doi: 10.1039/c6mb00168h.

2) Perez CJ, Monitoring Toxic Ionic Liquids in Zebrafish (Danio rerio) with Desorption Electrospray Ionization Mass Spectrometry Imaging (DESI-MSI). J Am Soc Mass Spectrom. 2017 Jun;28(6):1136-1148. doi: 10.1007/s13361-016-1515-9.

3) Zhao X, Huang X, Zhang X, Shi J, Jia X, Zhu K, Shao B. Distribution visualization of the chlorinated disinfection byproduct of diazepam in zebrafish with desorption electrospray ionization mass spectrometry imaging.Talanta. 2022 Jan 15;237:122919. doi: 10.1016/j.talanta.2021.122919.

Therefore, the authors should make an effort to improve their literature review.

  • We accept the comment about adding references regarding changes in the endogenous metabolome distribution upon exposure to toxicants and have edited the text to cite the work by Pirro et al., as well as an additional paper. However, we purposely avoided including additional literature regarding spatial mapping of toxicants to keep the introduction in line with the focus of our work, i.e., endogenous lipidome. We do include a single reference to work involving mapping toxicants in Daphnia because this is the only true MSI application in Daphnia, but we explicitly state this is not the focus of the paper - “While not a study of endogenous metabolism”

Moreover, the authors made a huge effort in explaining the evolvement  of D. magnain biological testing, but almost nothing was written about the hazard of bisphenol A (they wrote a couple of sentences at lines 416-417). They should emphasize this point in the introduction, too.

  • We had opted to limit the text pertaining to BPA as the focus of the paper is on the DESI-MS workflow and its potential applications in toxicology, where the model toxicant is a case study to evidence this. However, as requested, we have added some further information.

Line 444. They should also explain in the introduction why they choose to expose the  D. magna to such high concentrations of bisphenol A(5, 8, 10, 12, 20 ppm).  Are they reasonable concentrations easily encountered in real life? I do not think so. Are there any protocols suggesting such high concentrations for testing biological changes due to hazards expositions? Please provide a suitable explanation.

  • As mentioned in the paper (line  311), we selected a BPA concentration to maximise “the likelihood of inducing measurable molecular effects by DESI-MS”. A reference has been added after this statement to a LC-MS experiment studying the lipidome upon BPA exposure of similar (in fact higher) concentrations. We agree that this is not an environmentally relevant concentration, but investigating the environmental impacts of BPA was not a specific objective. 

Line 444 Please change ppm to mg/kg.

  • This change has been applied across all occurrences of ‘ppm’ in terms of BPA concentration in the main text.

I wonder if the authors were able to detect/map bisphenol A in the samples.

  • We had unsuccessfully attempted this, concluding that the concentration of BPA in the samples was below the LOD of the system. We have now included this text in the main paper, lines 347-350.

Line 360 ESI LC MS does not exist. The authors meant LC-ESI-MS, I guess.

  • We thank the reviewer for spotting this error and have corrected it as suggested,

Round 2

Reviewer 1 Report

 I am satisfied with the revised manuscript.